# *Toxoplasma gondii* virulence factor ROP1 reduces parasite susceptibility to murine and human innate immune restriction

**Simon Butterworth[1], Francesca Torelli[1], Eloise J. Lockyer[1], Jeanette Wagener[1], Ok-Ryul Song[2], Malgorzata Broncel[1,3], Matt R. G. Russell[4], Aline Cristina A. Moreira-Souza[1], Joanna C. Young[1], Moritz Treeck [1] ***

**1** Signalling In Apicomplexan Parasites Laboratory, The Francis Crick Institute, London, United Kingdom, **2** High-Throughput Screening Science Technology Platform, The Francis Crick Institute, London, United Kingdom, **3** Proteomics Science Technology Platform, The Francis Crick Institute, London, United Kingdom, **4** Electron Microscopy Science Technology Platform, The Francis Crick Institute, London, United Kingdom

\* moritz.treeck@crick.ac.uk

**Data Availability Statement:** Mass spectrometry acquisition files and MaxQuant processing files have been deposited to the ProteomeXchange

## Abstract

*Toxoplasma gondii* is an intracellular parasite that can infect many host species and is a cause of significant human morbidity worldwide. *T. gondii* secretes a diverse array of effector proteins into the host cell which are critical for infection. The vast majority of these secreted proteins have no predicted functional domains and remain uncharacterised. Here, we carried out a pooled CRISPR knockout screen in the *T. gondii* Prugniaud strain *in vivo* to identify secreted proteins that contribute to parasite immune evasion in the host. We demonstrate that ROP1, the first-identified rhoptry protein of *T. gondii*, is essential for virulence and has a previously unrecognised role in parasite resistance to interferon gamma-mediated innate immune restriction. This function is conserved in the highly virulent RH strain of *T. gondii* and contributes to parasite growth in both murine and human macrophages. While ROP1 affects the morphology of rhoptries, from where the protein is secreted, it does not affect rhoptry secretion. Finally, we show that ROP1 co-immunoprecipitates with the host cell protein C1QBP, an emerging regulator of innate immune signaling. In summary, we identify putative *in vivo* virulence factors in the *T. gondii* Prugniaud strain and show that ROP1 is an important and previously overlooked effector protein that counteracts both murine and human innate immunity.

## Author summary

*Toxoplasma gondii* is a single-celled eukaryotic pathogen that can infect many different species, including mice and humans. *T. gondii* secretes a large number of proteins into host cells that it infects, although the majority of these proteins are not well studied. We have carried out a knockout screen to identify *T. gondii* genes that are important for the parasite to survive during infection of a mouse. One of the genes we identified encodes the parasite protein ROP1, which was shown 30 years ago to be secreted into the host cell, but whose function remains unknown. We show that deletion of ROP1 causes an

Consortium via the PRIDE partner repository with the identifier PXD032319. All other data are within the manuscript and its Supporting Information files.

**Funding:** This work was supported by funding to MT from the Francis Crick Institute which receives its core funding from Cancer Research UK (FC001189), the UK Medical Research Council (FC001189), and the Wellcome Trust (FC001189). The Science Technology Platforms at the Francis Crick Institute receive funding from Cancer Research UK (FC001999), The UK Medical Research Council (FC001999) and the Wellcome Trust (FC001999). The funders had no role in study design, data collection and analysis, decision to publish, or preparation of the manuscript. For the purpose of Open Access, the authors have applied a CC BY public copyright licence to any Author Accepted Manuscript version arising from this submission.

**Competing interests:** The authors have declared that no competing interests exist.

otherwise lethal infection to be efficiently cleared by the host immune system. ROP1 is important for *T. gondii* to evade the cell autonomous immune responses of both human and murine cells, which is usual as the key mechanisms that control intracellular pathogens differ between humans and mice. ROP1 may interact with the host protein C1QBP, indicating the direction of future work to establish a mechanistic link to immune evasion.

## Introduction

*Toxoplasma gondii* is a single-celled intracellular parasite that is remarkable in its ability to infect any warm-blooded animal, including humans. In intermediate hosts, *T. gondii* tachyzoites must evade host immune clearance long enough to disseminate throughout the host organism and differentiate into the cyst-forming bradyzoites, which can be transmitted to the definitive feline host [1].

To this end, *T. gondii* secretes effector proteins into the host cell which modulate and counteract host innate immunity pathways [2,3]. These effector proteins are secreted from the rhoptries and dense granules, specialised secretory organelles found in the *Apicomplexa*.

Several effectors have previously been identified by quantitative trait locus (QTL) mapping following genetic crosses between strains of *T. gondii* with differing virulence in mouse models of infection [4–6]. This approach led to the discovery of ROP5 and ROP18, rhoptry proteins that cooperate to inhibit loading of host immune-related GTPases (IRGs) onto the parasitophorous vacuole membrane (PVM), and that are the major determinants of virulence in mice between different strains of *T. gondii* [7–12]. However, genetic cross approaches are limited in that they cannot identify effector proteins with the same function in both parental strains; for example, the dense granule protein GRA12, which has been shown to be a major virulence factor in both type I and type II laboratory strains of *T. gondii* [13,14].

Recently, we and others have used targeted, pooled CRISPR knockout screening to identify *T. gondii* genes which are required for survival and growth in mouse models of infection [15,16]. By comparison to *in vitro* growth phenotypes, it is possible to identify genes that are only required for parasite growth *in vivo*, and thus may have roles in evasion of the host immune response. These studies primarily targeted genes encoding proteins localised to the rhoptries and dense granules, as these proteins are secreted into the host cell and have the potential to interact with host proteins. However, 142 proteins that have only recently been localised to the rhoptries and dense granules have yet to be characterised [17].

To address this knowledge gap, we here use our previously described platform for customisable pooled CRISPR knockout screening in *T. gondii* [15] to screen an expanded library of rhoptry and dense granule protein-encoding genes for *in vivo* growth phenotypes in the type II Prugniaud (PRU) strain of *T. gondii*. We report phenotype scores for 164 genes, and identify ten putative virulence factors in addition to eight previously published virulence factors. These putative effectors include the prototypical rhoptry protein ROP1, whose function has been unknown to date. We demonstrate that ROP1 protects against IFNγ-mediated restriction in human and murine macrophages in both the PRU and RH strains of *T. gondii*. Although ROP1 affects rhoptry morphology, it does not affect rhoptry secretion; therefore, this role in suppression of innate immune restriction is an intrinsic function of ROP1. Knockout of ROP1 in the PRU strain of *T. gondii* renders these parasites avirulent in mice, confirming the important role of this secreted effector protein *in vivo*. Finally, we show that ROP1 co-immunoprecipitates with the host protein C1QBP.

## Results

### CRISPR screen

To screen for *T. gondii* effector proteins required for immune evasion, we cloned 906 proto-spacer sequences targeting 235 rhoptry and dense granule protein-encoding genes into a Cas9-sgRNA vector [15] (**S1 Data**). We transfected the resulting plasmid pool into the type II PRUΔHXGPRT strain of *T. gondii*, and selected for integration of the plasmids into the para-site genome for six days in human foreskin fibroblasts (HFFs) using a drug resistance marker. The surviving parasites were used to infect five C57BL/6J mice with 200,000 parasites each by injection into the peritoneum. After five days of infection, parasites were recovered from the mice by peritoneal lavage and expanded in HFFs for one passage (**Fig 1A**). To quantify the

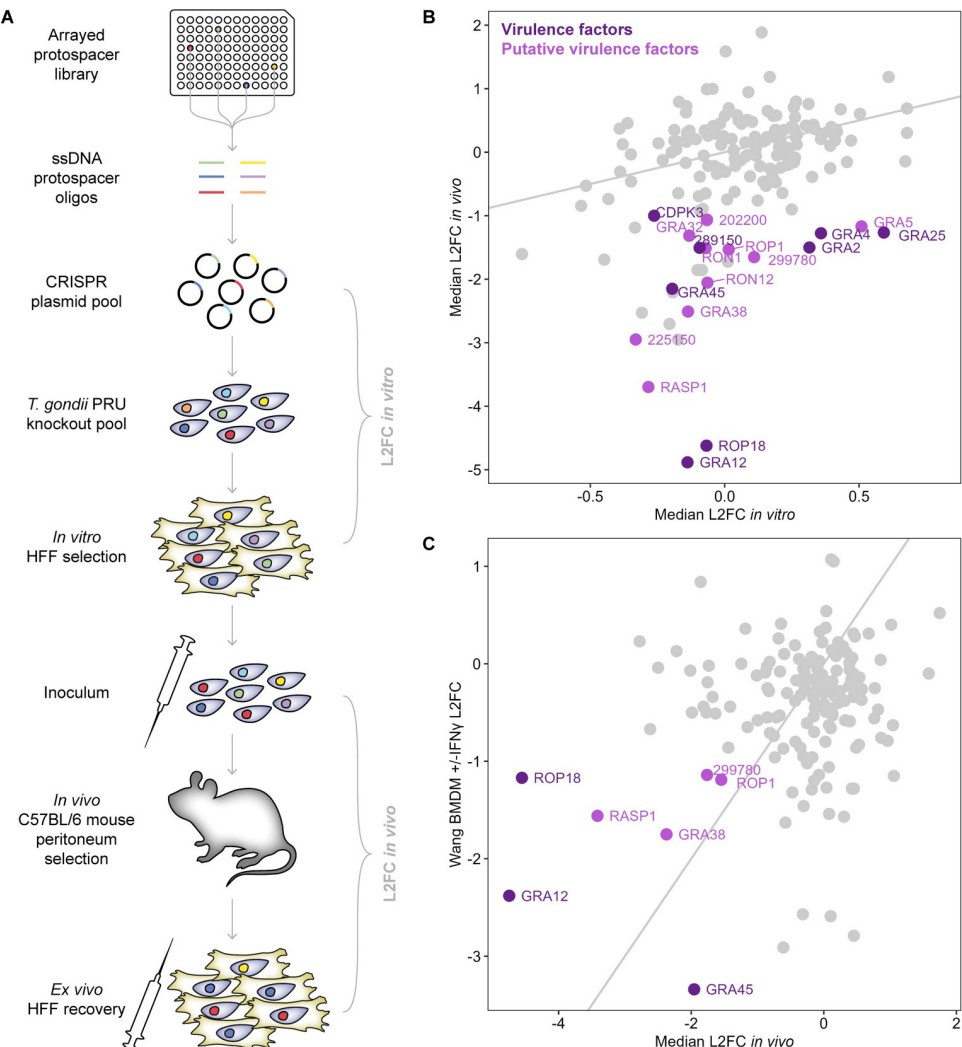

**Fig 1. Targeted *in vivo* CRISPR-Cas9 knockout screening of *T. gondii* rhoptry and dense granule proteins. A.** Schematic of knockout screen workflow. Protospacers encoded on arrayed ssDNA oligonucleotides are assembled by pooled Gibson cloning into a Cas9-sgRNA vector. The resulting plasmid pool is transfected into *T. gondii* PRU and the parasites selected *in vitro* in HFFs for integration for six days. Surviving parasites are used to infect five mice by intraperitoneal injection, recovered after five days and expanded for one further lytic cycle *in vitro*. The sgRNA cassettes are amplified from plasmid or parasite genomic DNA and sequenced to determine the relative abundance of each guide. **B.** Scatter plot of median L2FCs for each gene *in vitro* and *in vivo*. Putative virulence factors are identified as genes with an *in vivo* L2FC ≤ -1 and HFF z-score [18,19] ≥ -1 (**S1F Fig**). The grey line indicates equal L2FCs. **C.** Correlation between median L2FCs *in vivo* from this study and L2FCs between IFNγ-stimulated versus unstimulated BMDMs from [19]. Genes with a L2FC ≤ -1 in both screens are labelled. The grey line indicates equal L2FCs.

growth of parasite mutants in cell culture, sgRNAs were amplified by PCR from the plasmid pool and from genomic DNA extracted from the parasites after the *in vitro* drug selection. To quantify growth *in vivo*, sgRNAs were amplified from the leftover mouse inoculum and from the five recovered *ex vivo* populations. The sgRNAs from each population were sequenced by Illumina sequencing to determine their relative abundance.

To quantify the contribution of each gene to parasite growth, we calculated a phenotype score as the median $\log_2$-fold-change (L2FC) of the sgRNAs targeting a given gene during the *in vitro* and *in vivo* selections (drug-selected parasites vs. plasmid library and *ex vivo* population vs. inoculum respectively) (**Fig 1B**). We obtained such scores for 164 genes after filtering to remove genes with fewer than three well-represented sgRNAs (**S1 Data**). *In vivo* phenotype scores correlated strongly with those from our previous study (Young et al. 2019) (**S1A Fig**). *In vitro* phenotype scores also correlated well overall with our previous study and with two other genome-wide datasets from the RH strain of *T. gondii* [18,19] (**S1B–S1D Fig**), though they did not clearly separate some known essential genes (**S1E Fig**).

Therefore, to identify putative virulence factors which may have roles in host immune evasion, we selected genes with an *in vivo* median L2FC in this screen less than -1, then excluded genes with a *in vitro* HFF z-score less than -1 in either genome-wide dataset [18,19] (**Figs 1B and S1F**). This resulted in a set of 18 putative virulence factors, including seven that have previously been validated by single knockouts in Type II strains of *T. gondii* (GRA12 [13], ROP18 [20], GRA25 [21], GRA4 [13], TGME49_289150 [15], GRA2 [13], and CDPK3 [22]) and GRA45 which has been shown to affect virulence in the RH strain [19].

One surprising candidate among these hits was ROP1, as knockout of this gene was previously shown to have no effect on virulence in the RH strain [23]. ROP1 was not included in a recent *in vivo* screen in the RH strain [16], but was found to have a moderately negative, albeit not significant, fitness phenotype in IFNγ-stimulated versus naive C57BL/6J bone marrow-derived macrophages (BMDMs) [19]. ROP1 was one of only seven genes with an *in vivo* L2FC less than -1 in both this screen and in IFNγ-stimulated versus naive BMDMs, indicating that it may have a function in immune evasion in both the PRU and RH strains (**Fig 1C**). As the function of ROP1 has been unknown to date, aside from an interesting rhoptry morphology phenotype [23], we therefore decided to focus the remaining work on this gene. Given the fitness phenotypes described above, we hypothesised that ROP1 contributes to growth of parasites *in vivo* by inhibiting IFNγ-mediated restriction in infected macrophages, potentially through facilitating efficient secretion of rhoptry proteins into the host cell.

## ROP1 localises to the parasitophorous vacuole membrane up to 24 h post-invasion

To investigate the function of ROP1, we generated knockout cell lines in both the PRUΔKU80 and RHΔKU80 strains by replacing the coding sequence of ROP1 with an mCherry-T2A-HXGPRT drug selection cassette. We then complemented these lines with strain-matched ROP1-HA constructs integrated at the UPRT locus. Correct genomic integration of these constructs was verified by PCR, and the expected presence/absence of ROP1 was demonstrated by Western blot and immunofluorescence assay (IFA) (**Figs 2A and S2A–S2C**).

We noted that ROP1 is detectable by IFA at the parasitophorous vacuole membrane (PVM) up to at least 24 h post-invasion when the cells are permeabilised for a shorter period (1 min versus 15 min with 0.2% Triton X-100), which allows better visualisation of proteins at the PVM (**Figs 2B and S2C**). This contrasts with a previous report based on immuno-electron microscopy that ROP1 was present on the PVM immediately after invasion but not at 6 h

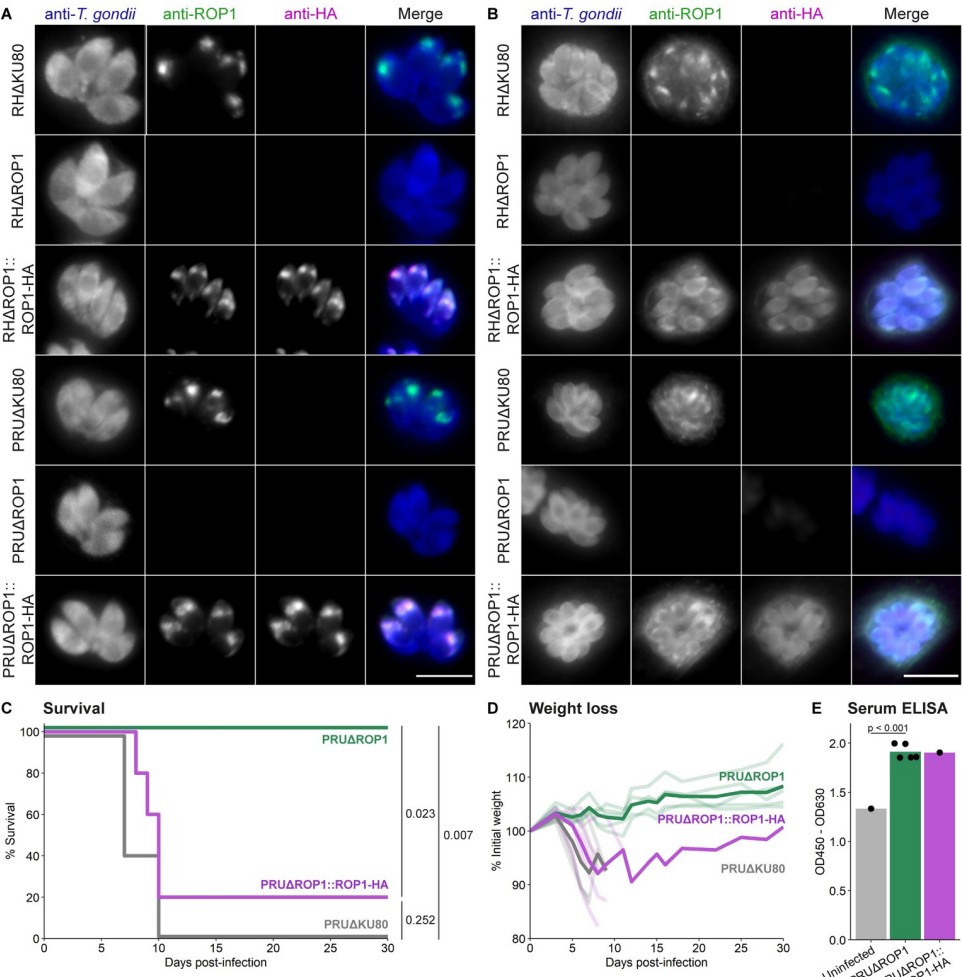

**Fig 2. PRUΔROP1 parasites are avirulent *in vivo*. A, B.** Immunofluorescence verification of ROP1 knockout and complemented *T. gondii* cell lines using **A** 15 minute permeabilisation or **B** 1 minute permeabilisation. Scale bar = 10 μm. **C.** C57BL/6J mice were infected with 20,000 parasites each by intraperitoneal injection and survival was monitored for 30 days. p-values were calculated by log-rank test with Benjamini-Hochberg adjustment; n = 5 mice per group. **D.** Weight loss of infected mice as a percentage of initial weight. Individual mice are shown as pale lines; dark lines represent mean weight loss of surviving mice. **E.** Serum ELISA of mice surviving at 30 days post-infection against *T. gondii* tachyzoite lysate antigen.

post-invasion [24], but is supported by a recent proximity biotinylation study which demonstrated that ROP1 is accessible to host cytosolic proteins at 24 h post-invasion [25].

Consistent with our CRISPR screen *in vitro* phenotype and a previous study [23], we did not see a major growth defect of the RHΔROP1 or PRUΔROP1 strains compared to the parental strains by plaque assay (**S2D Fig**), demonstrating that ROP1 is dispensable in the absence of immune pressure.

## PRUΔROP1 parasites are avirulent *in vivo*

ROP1 knockout has previously been shown to have no effect on RH strain virulence in Swiss Webster mice [23]. However, the RH strain is extremely virulent in most laboratory mouse strains, which can mask virulence phenotypes that are apparent in other *T. gondii* isolates, as, for example, in the case of Cyclase-Associated Protein [26]. We therefore wanted to determine

whether ROP1 knockout would affect the virulence of the PRU strain of *T. gondii in vivo*, as suggested by our CRISPR screen fitness phenotypes.

We infected C57BL/6J mice with 20,000 parasites each by intraperitoneal injection. While all the PRUΔKU80-infected mice succumbed to infection within 10 days, all the PRUΔROP1-infected mice survived for the full duration of the experiment (**Fig 2C**). Despite seroconverting against *T. gondii* tachyzoite lysate antigen, the PRUΔROP1-infected mice did not experience weight loss, one of the major clinical symptoms of acute toxoplasmosis in mice (**Fig 2D and 2E**). Virulence was rescued by complementation, though one PRUΔROP1:: ROP1-HA-infected mouse recovered and survived until the end of the experiment after initially losing weight. This surviving mouse may indicate incomplete rescue of function by the HA-tagged variant of ROP1, though it may also be that this mouse received a slightly lower inoculum. Regardless, these data confirm ROP1 as an important virulence factor *in vivo*.

## ROP1 contributes to *T. gondii* resistance to IFNγ-mediated restriction in murine and human macrophages

To test our hypothesis that ROP1 contributes to virulence by protecting the parasite against host IFNγ-mediated restriction, we quantified *T. gondii* growth in primary bone marrow-derived macrophages (BMDMs). BMDMs in 96-well plates were stimulated with IFNγ for 24 h or left unstimulated, then infected with mCherry-expressing parasite strains at an MOI of 0.3 for a further 24 h. Fluorescence microscopy images of the infected cells were captured using a high-content imaging system and the number of parasites, vacuoles, and host cells determined. The percentage survival of the parasites in IFNγ-stimulated cells was calculated relative to unstimulated cells (**S2 Data**).

Using total parasite number as an indicator of overall IFNγ-dependent restriction, we found that both the RHΔROP1 and PRUΔROP1 strains were significantly more restricted compared to UPRT knockout controls, and that this was rescued by complementation (**Fig 3A**). There was substantial variability in the apparent strength of restriction between replicates, potentially as a result of variable IFNγ-dependent host cell loss between different batches of BMDMs (**S3B Fig**). Nonetheless, consistent with our CRISPR screen fitness scores, PRUΔROP1 had an intermediate IFNγ resistance phenotype compared to PRUΔGRA12, a virulence factor which has been shown to be highly susceptible to IFNγ-mediated restriction in murine cells [13]. We found that the number of vacuoles was significantly reduced for RHΔROP1 and PRUΔROP1 compared to the control parasite lines, but did not find differences in vacuole size (**Figs 3B and S3A**), indicating that ROP1 knockout parasites are more susceptible to clearance through vacuole disruption, host cell death, or early parasite egress. In contrast, for PRUΔGRA12 we found both significantly reduced vacuole number and reduced vacuole size, indicating an alternative or additional mechanism of restriction.

The reduction in vacuole number for ΔROP1 parasites suggested increased susceptibility to vacuole disruption by the immune-related GTPases (IRGs), as found for the strain-dependent rhoptry effectors ROP5, ROP17 and ROP18 [7–12]. We were therefore interested in determining whether ROP1 might contribute to parasite survival in human macrophages, which also restrict parasite growth in an IFNγ-dependent manner but lack the IRG system responsible for restriction in murine cells [3]. We quantified parasite growth in IFNγ-stimulated THP-1-derived macrophages by high-content imaging as above (**S3 Data**), and found, surprisingly, that RHΔROP1 and PRUΔROP1 parasites were more restricted than either control or complemented parasite lines (**Fig 3C**). For PRUΔROP1 we observed the same reduction in vacuole number but not in vacuole size as in BMDMs, although for RHΔROP1 we found a significant decrease in the number of vacuoles compared to the complemented line but not compared to RHΔUPRT, and instead observed a modest but significant decrease in vacuole size compared

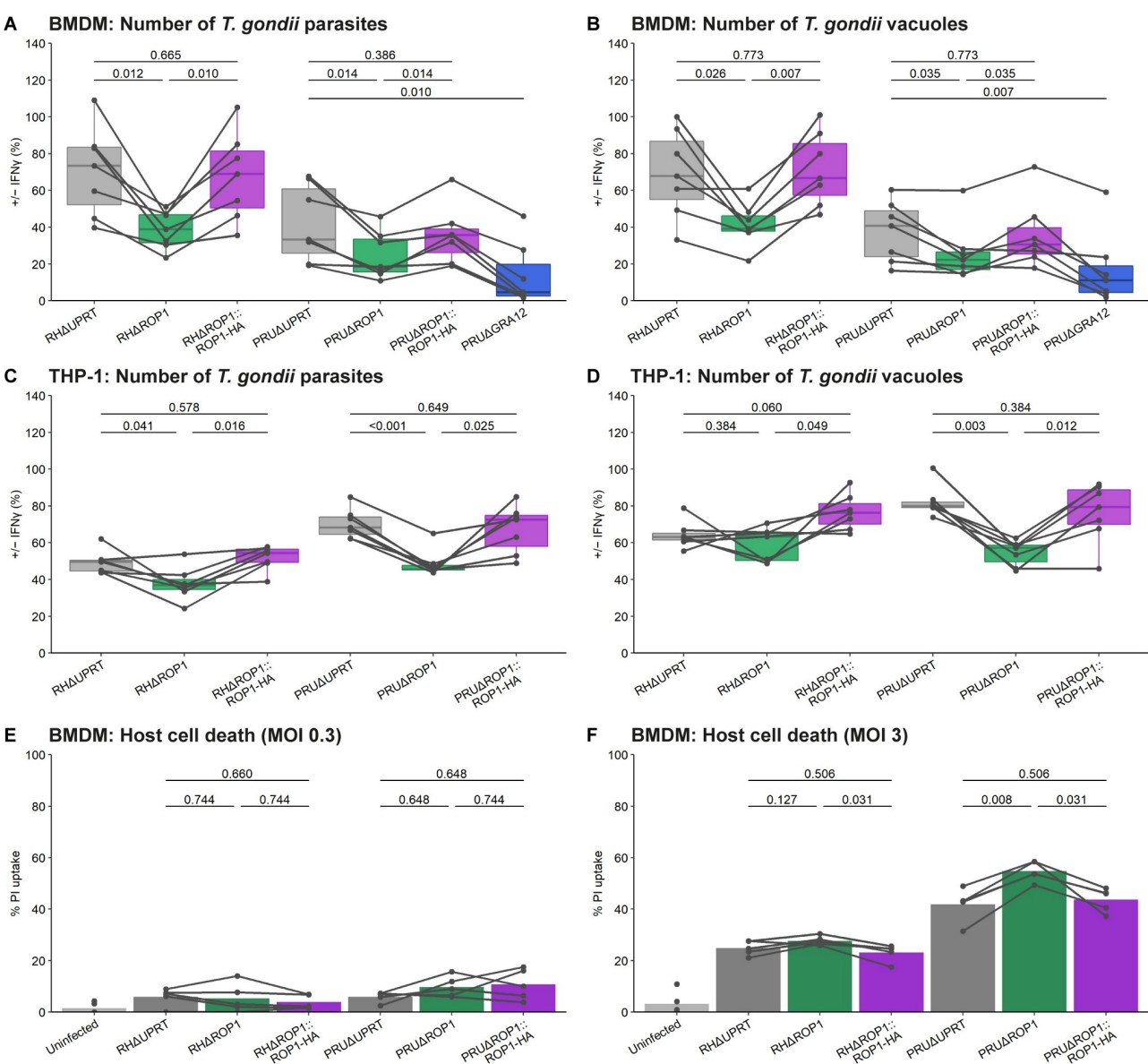

**Fig 3. ROP1 contributes to *T. gondii* resistance to IFNγ in murine and human macrophages. A, B.** *T. gondii* growth restriction in IFNγ-stimulated BMDMs infected at an MOI of 0.3 for 24 h, quantified by high-content imaging and automated analysis. Parasite growth in IFNγ-stimulated macrophages is shown as a percentage of that in unstimulated macrophages in terms of **A** total parasite number and **B** vacuole number. p-values were calculated by paired two-sided *t*-test with Benjamini-Hochberg adjustment. **C, D.** *T. gondii* growth restriction in IFNγ-stimulated THP-1-derived macrophages infected at an MOI of 0.3 for 24 h, quantified by high-content imaging and automated analysis. Parasite growth in IFNγ-stimulated macrophages is shown as a percentage of that in unstimulated macrophages in terms of **A** total parasite number and **B** vacuole number. p-values were calculated by paired two-sided *t*-test with Benjamini-Hochberg adjustment. **E. F.** Propidium iodide uptake of IFNγ-stimulated BMDMs infected at an MOI of **E** 0.3 or **F** 3 for 12 h. p-values were calculated by paired two-sided *t*-test with Benjamini-Hochberg adjustment.

to RHΔUPRT, which was not rescued by complementation (**Figs 3D and S3C**). These results therefore indicate that the mechanism of restriction of ΔROP1 parasites is independent of the IRGs and that instead ROP1 targets a conserved mechanism of restriction in both murine and human macrophages, although it is also possible that ROP1 targets different mechanisms in different parasite strain and host cell contexts.

Host cell death has been shown to play an important role in IFNγ-mediated parasite restriction in both murine and human macrophages infected at a high MOI, although there is

typically little detectable cell death in cells infected at MOI < 1 [27–29]. To test whether ROP1 affects host cell death, we measured propidium iodide uptake over time in infected BMDMs, which indicates host cell membrane permeabilisation. At an MOI of 0.3, as targeted in our IFNγ restriction assays, we did not find any significant differences in propidium iodide uptake between the different parasite strains (**Figs 3E and S4A**). In contrast, at an MOI of 3 we found that the PRUΔROP1 strain induced significantly higher levels of host cell death compared to PRUΔUPRT at 12 hours post-infection, which was rescued by complementation (**Figs 3F and S4B**). A similar trend was apparent for RHΔROP1 compared to the complemented line, but was not significant compared to RHΔUPRT. In both murine and human macrophages, host cell death at high MOI appears to result from exposure of parasite material to host cytosolic sensors following disruption of the parasitophorous vacuole [27–30]. Our data together from both the IFNγ restriction assays and host cell death measurements could therefore indicate a role for ROP1 in preventing vacuole disruption, which at high MOI has the secondary effect of reducing parasite-induced host cell death, particularly for the PRU strain. Alternatively, ROP1 may directly inhibit host cell death pathways or prevent early parasite egress, which results in host cell death through rupture of the host cell plasma membrane.

## ROP1 affects rhoptry morphology, but not ROP secretion

Knockout of ROP1 in the RH strain has previously been shown to affect the ultrastructure of the rhoptries: wild-type rhoptries have a heterogeneous texture by transmission electron microscopy, whereas in the absence of ROP1 the rhoptries show a homogeneous, electron-dense structure [23]. We were able to reproduce this phenotype previously observed in the RH strain, and additionally show that it is conserved in the type II PRU strain (**Figs 4A and S5**). This morphological change suggested that ROP1 might promote parasite survival in both murine and human macrophages by facilitating the secretion of other rhoptry proteins with host- or parasite strain-specific roles in counteracting IFNγ-mediated restriction. RHΔROP1 parasites have been shown to have the same invasion rate as the parental strain [23], indicating normal secretion of the rhoptry neck (RON) proteins with critical roles in host cell invasion [31]; however, given the altered morphology of the rhoptries and rhoptry bulb localisation of ROP1, we hypothesised that secretion of other rhoptry bulb (ROP) proteins might be affected by ROP1 knockout independently of the RON proteins.

As a proxy for secretion, we analysed host cell phenotypes induced by three ROP proteins: phosphorylation of STAT6 by ROP16 (in the RH strain but not PRU) [32], ROP17-dependent induction of cMyc expression in the host nucleus [33], and recruitment of Irgb6 to the parasitophorous vacuole membrane, which is inhibited by ROP18 in the RH strain [7] (**Fig 4B–4D**). We did not find a significant difference between ΔROP1 and parental lines in any of these assays, although we observed the expected decrease in host STAT6 phosphorylation for PRUΔKU80 versus RHΔKU80-infected cells, decrease in host cMyc expression for PRUΔMYR1 versus PRUΔKU80-infected cells, and increase in Irgb6 recruitment to RHΔROP18 versus RHΔKU80 vacuoles. It is therefore unlikely that ROP1 has a major function in secretion of ROP proteins. The functional consequences, if any, of the altered rhoptry morphology remain unknown, as neither RON nor ROP secretion is affected. These experiments instead suggest that resistance to IFNγ-mediated restriction is most likely an inherent function of ROP1.

## ROP1 co-immunoprecipitates with host C1QBP

To identify interacting partners of ROP1 that might inform on its function, we generated cell lines in which ROP1 was tagged at the endogenous C-terminus with a single haemagglutinin (HA) epitope. Correct integration of the tagging construct into the genome was confirmed by

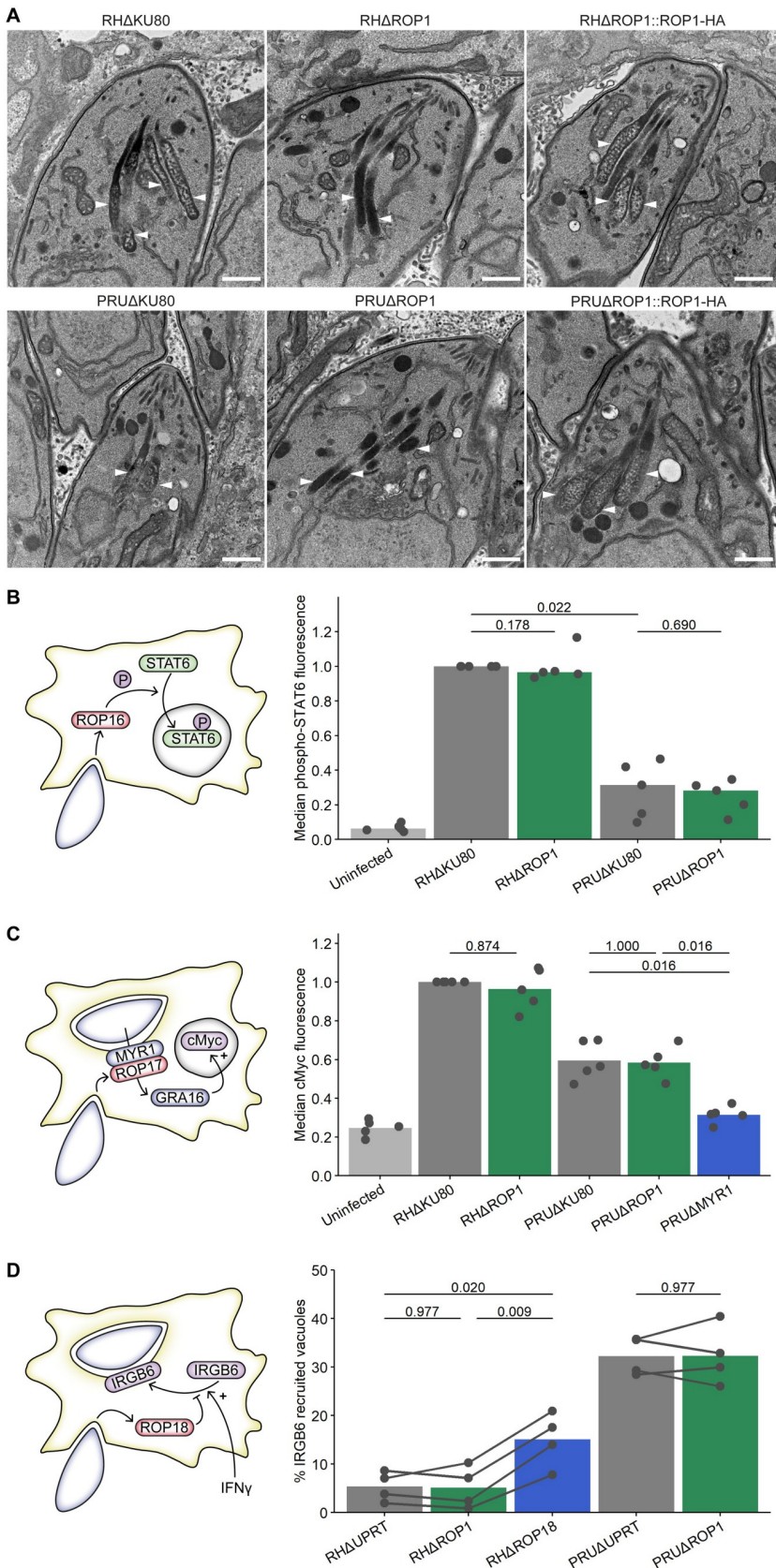

**Fig 4. ROP1 knockout alters rhoptry morphology but not ROP secretion. A.** TEM images of the rhoptries of intracellular tachyzoites. White arrowheads indicate rhoptries. Scale bar = 500 μm. **B.** Normalised median anti-phospho-STAT6 fluorescence intensity of infected HFFs. HFFs were infected for 2 h, fixed with methanol and stained with anti-phospho-STAT6 and anti-*T. gondii*, then analysed by flow cytometry. p-values were calculated by two-sided Wilcoxon rank sum test with Benjamini-Hochberg adjustment. **C.** Normalised median nuclear anti-cMyc fluorescence intensity of infected HFFs. HFFs were infected for 24 h in 0.1% FBS medium, fixed and stained with anti-cMyc and anti-*T. gondii*, and the median nuclear anti-cMyc fluorescence intensity was determined from immunofluorescence microscopy images. p-values were calculated by two-sided Wilcoxon rank sum test with Benjamini-Hochberg adjustment. **D.** Recruitment of host IRGB6 to *T. gondii* vacuoles in BMDMs. BMDMs were stimulated with IFNγ for 24 h, infected with *T. gondii* cell lines for 1 h, fixed and stained with anti-IRGB6. The percentage of vacuoles decorated with IRGB6 was determined by automated fluorescence imaging and analysis. p-values were calculated by paired two-sided *t*-test with Benjamini-Hochberg adjustment.

PCR (**S6A Fig**), expression of ROP1-HA was demonstrated by Western blot (**S6A Fig**), and correct localisation of ROP1-HA was determined by IFA (**Fig 5A**).

We carried out anti-HA immunoprecipitation in IFNγ-stimulated primary C57BL/6J murine embryonic fibroblasts (MEFs) infected with either PRU ROP1-HA or parental PRUΔKU80 at 24 hours post-infection. We used MEFs for this experiment due to the difficulty of obtaining large enough quantities of BMDMs, and because MEFs are thought to restrict *T. gondii* through the same mechanisms as BMDMs [34]. Immunoprecipitated proteins were in-gel digested and identified and quantified by liquid chromatography-tandem mass spectrometry (**S4 Data**).

As expected, ROP1 was strongly enriched in the PRU ROP1-HA-infected samples but not in the PRUΔKU80-infected samples (**Fig 5B**). Aside from ROP1, all significantly enriched proteins were host rather than *T. gondii* proteins. Among the most highly enriched proteins were a ubiquitin-conjugating enzyme, RNF2, and a deubiquitinating enzyme, USP17LA. Therefore, we hypothesised that ROP1 may interfere with ubiquitination of the vacuole, which normally serves to recruit the guanylate-binding protein (GBP) GTPase restriction factors to the vacuole [35]. However, we did not observe any difference in the percentage of ubiquitin-decorated vacuoles of ΔROP1 and parental parasites at 3 hours post-infection in BMDMs (**S7 Fig**), suggesting that ROP1 does not impact this mechanism of restriction.

Instead, we focused on the most strongly enriched host protein in the PRU ROP1-HA samples versus PRUΔKU80: Complement Component 1q Binding Protein (C1QBP, also known as GC1QR, HABP1, p32, p33, SF2P32) (**Fig 5B**). C1QBP is a small acidic protein which forms a homotrimer with a highly asymmetric charge distribution [36]. Intriguingly, C1QBP has been implicated as a regulator of autophagy and innate immune signalling, in addition to diverse other functions in different cellular compartments [37–41].

To validate the co-immunoprecipitation of ROP1 with C1QBP, we repeated the anti-HA immunoprecipitation using both the RH and PRU strains in MEFs and HFFs and checked for enrichment of C1QBP by Western blot. We saw that in all combinations ROP1-HA pulled down C1QBP, while an unrelated HA-tagged rhoptry protein TGME49_202200 [17] did not (**Fig 5C and 5D**).

We observed that C1QBP localised primarily to the mitochondria in MEFs, HFFs and BMDMs by co-staining with MitoTracker (**S8 Fig**). The specificity of this staining was validated in C1QBP$^{-/-}$ MEFs [41] (**S9 Fig**). We did not observe noticeable re-localisation of C1QBP following IFNγ stimulation in BMDMs (**S8A Fig**). Type I strains of *T. gondii*, but not type II strains, recruit host cell mitochondria to the parasitophorous vacuole via interaction of the parasite secreted protein MAF1B with components of the host mitochondrial protein import machinery and mitochondrial intermembrane space bridging complex [42–45]. Upon co-staining for ROP1 and C1QBP, we identified some co-localisation at the parasitophorous vacuole membrane of RHΔKU80 parasites (**Fig 5E and 5F**). This was not observed for

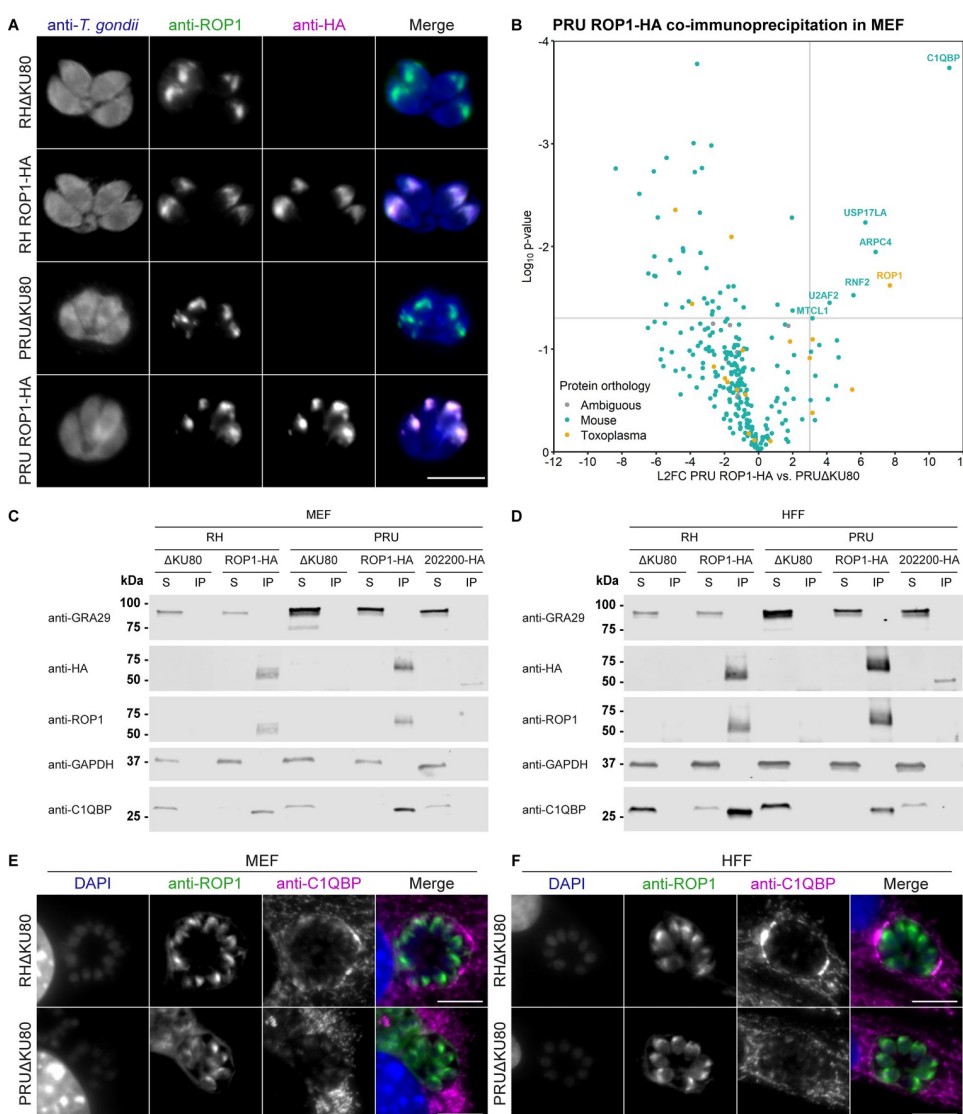

**Fig 5. ROP1 co-immunoprecipitates with host C1QBP. A.** Immunofluorescence verification of C-terminal HA-tagging of ROP1. Scale bar = 10 μm. **B.** Enrichment of proteins that co-immunoprecipitate with ROP1. Primary MEFs were infected with PRUΔKU80 or PRU ROP1-HA parasites for 24 h, following which ROP1 was immunoprecipitated with anti-HA agarose matrix and co-immunoprecipitated proteins were identified and quantified by mass spectrometry. L2FCs were calculated from the geometric mean of the iBAQ intensities across replicates, and p-values calculated by two-sided Welch's *t*-test. Proteins with p-value < 0.05 and L2FC > 3 are annotated. **C, D.** Co-immunoprecipitation of C1QBP with ROP1 in **C** primary MEFs and **D** HFFs infected with RH ROP1-HA, PRU ROP1-HA, and PRU 202200-HA. S = post-immunoprecipitation supernatant, IP = immunoprecipitate. Note that the immunoprecipitate fraction represents 3x the relative amount of the total lysate compared to the post-IP supernatant fraction. **E, F.** Immunofluorescence localisation of ROP1 and C1QBP in RHΔKU80 and PRUΔKU80 infected **E** primary MEFs and **F** HFFs at 24 hours post-infection. Scale bar = 10 μm.

PRUΔKU80 parasites that do not recruit host mitochondria. Although one report has indicated C1QBP can be recruited from a cytosolic pool to the outer membrane of the mitochondria upon RNA virus infection [38], others have indicated that C1QBP is exclusively localised to the mitochondrial matrix [46]. It is therefore unclear whether C1QBP would be able to interact with a parasite protein at the PVM. We attempted to validate the interaction of ROP1 with C1QBP *in cellulo* by proximity biotinylation; however, we found that C-terminal tagging

or ROP1 with the TurboID biotin ligase prevented secretion and localisation of ROP1 to the PVM (not shown).

We hypothesised that if the putative interaction with C1QBP was required for ROP1 to suppress IFNγ-dependent restriction, then ablation of C1QBP in the host cell would phenocopy ROP1 knockout in the parasites. We therefore carried out high-content imaging assays to measure IFNγ-dependent restriction of ΔUPRT and ΔROP1 parasites in immortalised MEFs derived from homozygous C1QBP$^{flox/flox}$ C57BL/6 mice in which C1QBP had been excised by transient transfection with Cre recombinase [41]. Although wild-type MEFs have previously been found to restrict both RH and PRU parasites [11,47], neither the C1QBP$^{flox/flox}$ nor C1QBP$^{-/-}$ MEFs were able to restrict parasite growth upon the addition of IFNγ (**S9 Fig**). We were therefore unable to draw conclusions regarding a possible role of C1QBP in parasite restriction in these cells.

## Discussion

We screened an expanded library of rhoptry and dense granule protein-encoding genes and identified 18 genes which contribute to *T. gondii* PRU growth *in vivo* in the mouse peritoneum, but not *in vitro*, indicating that these secreted effectors may be involved in evasion of host immune responses. Of the 235 targeted genes in this screen, we were able to determine phenotype scores for 164 genes with high confidence. This is because some protospacers had low read counts at the start of the experiment and dropped below the limit of detection over the course of the experiment. Future optimisation of CRISPR Cas9-sgRNA library preparation to achieve equal guide representation in the knockout vector pool could help to minimise drop-outs in genetic screens.

A key advantage of pooled CRISPR knockout screening is that it enables identification of virulence factors with the same function in many or all *T. gondii* strains, in contrast to genetic crosses which have been the major approach in the field until recently. By comparison to a recently published genome-wide dataset of CRISPR knockout phenotypes of the RH strain of *T. gondii* in IFNγ-stimulated BMDMs, we were able to identify a subset of effector proteins which are apparently important for immune evasion of both the PRU and RH strains, which would have been missed by genetic crosses between these strains. This comparison is likely imperfect, as different experimental models and protospacer libraries were used. Therefore, knockout screens which directly compare different strains of *T. gondii* in the same system with the same library will be an important area for future research and will provide a valuable resource to the community. Nevertheless, further study of the putative effectors important for RH and PRU infections of mice identified here, such as RASP1, GRA38, and TGME49_299780, may reveal new mechanisms of parasite virulence and subversion of the host cell.

In this work, we chose to focus on ROP1, the first-identified rhoptry protein of *T. gondii* whose function has remained mysterious for 30 years [24,48,49]. ROP1 was a somewhat-surprising hit in our screen, as this protein was previously shown to have no effect on virulence in the RH strain [23]. We showed in contrast that ROP1 is essential for virulence of the PRU strain. In both RH and PRU, ROP1 contributes to parasite resistance to IFNγ-mediated innate immune restriction in macrophages, likely explaining the virulence phenotype we observed as macrophages are the most commonly infected cell type in acute infection in the peritoneum [50] and IFNγ is the principle cytokine required for control of acute *T. gondii* infection *in vivo* [51]. It is likely that the very high virulence of the RH strain masked this role of ROP1 *in vivo* in prior work, highlighting the value of quantitative assessment of *in vivo* fitness in a less virulent *T. gondii* strain.

Interestingly, we found that ROP1 contributes to resistance to IFNγ-mediated restriction in both murine and human macrophages. This suggests that ROP1 counteracts an innate immune restriction mechanism which is common to both host species, although we cannot rule out a pleiotropic effect. ROP1 is protective in pre-activated immune cells, in contrast to the secreted effector IST that has been shown to protect against parasite restriction in THP-1 macrophages when interferon stimulation occurs after infection, but that is likely not protective when the host cells are pre-activated [52]. To our knowledge, the dense granule chaperone GRA45 is the only secreted protein other than ROP1 known to protect against *T. gondii* clearance in human macrophages that have been pre-activated with IFNγ [19].

The mechanism through which ROP1 protects against IFNγ-mediated clearance remains unclear. In our high-content imaging assays, we find reduced vacuole number as the major driver of restriction, and in our propidium iodide uptake assays we find an MOI-dependent increase in host cell death for ΔROP1 parasites. It is possible that ROP1 directly inhibits host cell death in some manner, or that ΔROP1 parasites egress early from the host cell at a higher rate than wild-type parasites, rupturing the host cell plasma membrane in the process. However, the moderate increase in cell death does not seem able to fully account for the ~40% relative reduction in parasite survival for both RHΔROP1 and PRUΔROP1 in BMDMs. An alternative explanation may be that the host cell death observed here is a secondary effect of vacuole disruption that releases parasite molecules into the cytoplasm to trigger cell death pathways, as has been documented in both BMDMs and THP-1-derived macrophages [27,29].

Previously observed ultrastructural changes in the rhoptries of ΔROP1 parasites, which we also observed, suggested that ROP1 may have a structural role in rhoptry function or in secretion into the host cell, and could thereby affect parasite restriction by altering the secretion of other rhoptry proteins [23]. However, we could not find any evidence that knockout of ROP1 affects the secretion of other rhoptry proteins, which concords with the lack of an invasion or *in vitro* growth phenotype [23]. Whether this ultrastructural change in the rhoptries upon deletion of ROP1 relates to the IFNγ restriction phenotype is unclear; the evidence nonetheless suggests that suppression of restriction is an intrinsic function of ROP1.

ROP1 from both RH and PRU parasites co-immunoprecipitates reliably with a host protein, C1QBP, from infections in both mouse and human cells. ROP1 has been observed in evacuoles (rhoptry-derived secretory vesicles) in the host cell cytoplasm immediately after secretion [53], and we observed ROP1 present at the present at the parasitophorous vacuole membrane up to at least 24 hours post-infection. The localisation of C1QBP is a matter of some debate and it is unclear whether an interaction between ROP1 and C1QBP *in cellulo* is topologically possible. Some reports have found that C1QBP is exclusively localised to the mitochondrial matrix, while others find an additional cytoplasmic pool of C1QBP protein which can be recruited to the outer membrane of the mitochondria upon infection with an RNA virus [38,46]. We attempted to validate an interaction of ROP1 with C1QBP by proximity biotinylation; however, tagging of ROP1 with the TurboID biotin ligase prevented secretion and localisation of ROP1 to the PVM, in contrast to the single HA tag that we used for co-immunoprecipitation. We therefore cannot rule out that the interaction observed between ROP1 and C1QBP is an artefact of cell lysis. Both proteins are noted for highly asymmetric charge distributions which could provide the basis for any (real or artefactual) interaction: the N-terminal region of ROP1 is highly acidic and the C-terminal region is highly basic [49], while acidic charge is highly concentrated on one face of the C1QBP trimer [36]. The asymmetric charge distribution of ROP1 was noted by Ossorio, Schwartzman, and Boothroyd to putatively facilitate interaction with host cell components [49].

We attempted to link C1QBP to IFNγ-dependent parasite restriction; however, the immortalised C1QBP$^{flox/flox}$ and C1QBP$^{-/-}$ MEFs we obtained were unable to restrict *T. gondii*

growth. Possibly, the immortalisation procedure or passage history of these cell lines has caused them to lose the ability to restrict *T. gondii*, as wild-type MEFs have otherwise previously been shown to restrict both the RH and PRU strains [11,47]. C1QBP knockout is lethal in mice, therefore it is not possible to obtain primary C1QBP$^{-/-}$ BMDMs [41]. We also attempted siRNA-mediated knockdown of C1QBP in wild-type primary BMDMs using a pool of three different siRNAs, but were only able to achieve <10% knockdown of C1QBP (not shown).

Nonetheless, C1QBP is an interesting candidate as host partner of a parasite virulence factor. C1QBP acts as a positive regulator of autophagy and mitophagy through stabilisation of ULK1 [37]; in both human and murine cells, autophagy proteins play critical roles in IFNγ-dependent parasite restriction [34], therefore modulation of this role of C1QBP as a regulator of autophagy could conceivably affect parasite restriction. Additionally, a growing body of evidence implicates C1QBP as a negative regulator of antiviral innate immunity pathways [38,54], and a number of viral proteins have been found to interact with C1QBP [39,55–57].

In summary, our data show that ROP1 is an important *T. gondii* effector protein that suppresses IFNγ-mediated restriction in macrophages and thereby contributes to parasite virulence. This function was previously overlooked in the RH strain, highlighting that further systematic study of parasite effectors in different strains and host cell types will likely reveal additional mechanisms of *T. gondii* immune evasion.

## Methods

### Ethics statement

All mouse work was approved by the UK Home Office (project license PDE274B7D) and the Francis Crick Institute Ethical Review Panel and carried out in accordance with the UK Animals (Scientific Procedure) Act 1986 and European Union directive 2010/63/EU.

**Mice.** C57BL/6J mice were bred and housed in pathogen-free conditions at the Biological Research Facility of the Francis Crick Institute.

**Cell culture.** All cell lines were cultured at 37˚C and 5% $CO_2$, and were tested monthly for *Mycoplasma spp.* contamination by PCR.

**HFF.** Primary HFFs (ATCC) were cultured in Dulbecco's Modified Eagle's Medium (DMEM) with 4.5 g/L glucose and GlutaMAX (Gibco) supplemented with 10% v/v heat-inactivated foetal bovine serum (FBS) (Gibco).

**BMDM.** Monocytes were isolated from the femurs of 6–12 week-old male C57BL/6J mice and differentiated into BMDMs for six days in 70% v/v RPMI 1640 medium (ATCC modification) (Gibco), 20% v/v L929 cell conditioned medium (provided by the Cell Services Science Technology Platform at the Francis Crick Institute), 10% v/v heat-inactivated FBS (Gibco), 100 U/mL penicillin-streptomycin (Gibco) and 50 μM 2-mercaptoethanol (Sigma). Following differentiation, BMDMs were cultured in the same medium without 2-mercaptoethanol.

**THP-1.** THP-1 cells were cultured in RPMI 1640 medium (Gibco) supplemented with 10% v/v FBS (Gibco). THP-1 monocytes were differentiated into macrophages with 100 ng/mL phorbol 12-myristate 13-acetate (Sigma) for 24 h, followed by a rest period without phorbol 12-myristate 13-acetate for 24 h.

**MEF.** Primary C57BL/6 MEFs (ATCC) and immortalised C57BL/6 C1QBP$^{flox/flox}$/C1QBP$^{-/-}$ MEFs (a gift from the lab of Dongchon Kang) were cultured in DMEM with 4.5 g/L glucose and GlutaMAX (Gibco) supplemented with 10% v/v heat-inactivated FBS (Gibco).

**Toxoplasma gondii.** All *T. gondii* tachyzoite cell lines were maintained by serial passage in HFFs. Parasites were harvested for experiments by mechanical lysis with a 27 G needle and passed through a 5 μm sterile filter. The parental lines used in this study were PRUΔHXGPRT

[58], RHΔKU80 [59], and PRUΔKU80 [60]. The genotypes of parasites used were verified by restriction fragment length polymorphism of the SAG3 gene [61].

## CRISPR screen

**Experimental protocol.** Pooled *in vivo* CRISPR knockout screening was performed as previously described [15]. Briefly, ssDNA oligonucleotides encoding protospacer sequences were selected from an arrayed library using an Echo 550 Acoustic Liquid Handler (Labcyte). Five protospacer sequences were selected per target gene, and dispensed in triplicate. The ssDNA oligonucleotides were integrated into a pCas9-GFP-T2A-HXGPRT::sgRNA vector [15] by pooled Gibson assembly.

The resulting plasmid pool was linearised and transfected into $10^7$ PRUΔHXGPRT tachyzoites using the Amaxa 4D Nucleofector system (Lonza) with buffer P3 and pulse code EO-115. After 24 h recovery, transfected parasites were selected in HFFs for integration of the plasmid into the genome with 25 μg/mL mycophenolic acid (Sigma) and 50 μg/mL xanthine (Sigma). Following selection, five mice were infected by intraperitoneal injection with 200,000 parasites each, as determined by plaque assay. After five days, parasites were recovered by peritoneal lavage and cultured in HFFs for one passage.

Genomic DNA was extracted from a sample of the parasite population following *in vitro* drug selection, from the leftover mouse inoculum, and from the five *ex vivo* populations using the DNeasy Blood and Tissue Kit (Qiagen). Illumina sequencing libraries were prepared by nested PCR amplification of the protospacer sequences from the parasite genomic DNA and the plasmid pool using primers 1–11. The libraries were sequenced on a HiSeq 4000 platform (Illumina) with 100 bp paired-end reads to a minimum depth of 7.5 million reads per sample (approximately 5000x coverage of the protospacer pool).

**Data analysis.** Following demultiplexing, the reads were trimmed and aligned to a reference of protospacer sequences using a custom perl script. Subsequent analysis was carried out using R v4.0.1 (https://www.r-project.org/) with packages tidyverse v1.3.1, qvalue v2.22.0, ggrepel v0.9.1 and scales v1.1.1.

Protospacers with fewer than 50 raw reads in every sample were removed from the analysis and remaining counts normalised using the median of ratios method [62]. Genes with fewer than three protospacers remaining were then removed from the analysis.

For each gene, the median *in vitro* L2FC was calculated from the normalised counts of the protospacers targeting that gene in the drug-selected parasite population and the plasmid pool. The median *in vivo* L2FC was calculated using the geometric mean of the normalised counts in the *ex vivo* parasite populations and the normalised counts in the inoculum. Genes in the top $5^{th}$ percentile of median absolute deviation of the *in vitro* or *in vivo* L2FCs were removed from the analysis.

In addition, for each gene an *in vitro* and *in vivo* p-value calculated by paired two-sided t-test on the $\log_2$-transformed normalised counts and adjusted to correct for local false discovery rate (FDR) using the qvalue R package. The median L2FCs and FDR-adjusted q-values were used to calculate a DISCO score for each gene as:

$$\text{abs} (\text{L2FC}_{\text{in vitro}} - \text{L2FC}_{\text{in vivo}}) * \text{abs} (\log_{10}(\text{q-value}_{\text{in vitro}}) + \log10(\text{q-value}_{\text{invivo}}))$$

## Generation of *T. gondii* cell lines

**Knockouts.** Inverse PCR was used to introduce a protospacer targeting the CDS of either ROP1 or UPRT to a pCas9-GFP::sgRNA plasmid using primers 12–14. For ROP1, A Pro$^{\text{GRA1}}$-mCherry-T2A-HXGPRT-Ter$^{\text{GRA2}}$ construct was amplified from a template plasmid [15] using primers 15 and 16 to induce 40 bp homology arms to the 5' and 3' UTRs of ROP1. For UPRT,

the above construct was amplified using primers 17 and 18 and integrated into BamHI/PacI-digested (NEB) pUPRT plasmid by Gibson assembly. 15 μg of homology repair template (purified PCR product or linearised pUPRT-mCherry-HXGPRT) was co-transfected with 15 μg pCas9 plasmid targeting the gene of interest into the RHΔKU80 and PRUΔKU80 lines using the Amaxa 4D Nucleofector system (Lonza) as above. After 24 h recovery, transfected parasites were selected with 25 μg/mL mycophenolic acid and 50 μg/mL xanthine for at least six days before single-cell cloning by serial dilution. Integration of the mCherry-HXGPRT cassette repair template was verified by PCR with primers 19–22.

**Complementation.** The ROP1 CDS together with 1000 bp upstream of the start codon was amplified from RHΔKU80 and PRUΔKU80 genomic DNA using primers 23–25. The backbone of the pUPRT plasmid was amplified with primers 26 and 27 and assembled with the ROP1 inserts by Gibson assembly. 15 μg of pUPRT-RH/PRU-ROP1-HA plasmid was linearised and transfected together with 15 μg pCas9 plasmid targeting UPRT into the RHΔROP1 and PRUΔROP1 lines. After 24 h recovery, transfected parasites were selected with 5 μM 5'-fluo-2'-deoxyuridine for at least six days before single-cell cloning by serial dilution. Integration of the pUPRT-RH/PRU-ROP1-HA plasmids into the UPRT locus was verified by PCR using primers 21 and 28.

**HA tagging.** Inverse PCR was used to introduce a protospacer targeting the 3' UTR of ROP1 into the pCas9-GFP::sgRNA plasmid using primers 12 and 29. Primers 30 and 31 were used to amplify an in-frame HA-Ter$^{GRA2}$::Pro$^{DHFR}$-HXGPRT-Ter$^{DHFR}$ construct from a template plasmid, introducing 40 bp homology arms to the 3' end of the ROP1 CDS. 15 μg each of pCas9 plasmid and purified PCR product were co-transfected as above into the RHΔKU80 and PRUΔKU80 lines. Selection with mycophenolic acid and xanthine and cloning were carried out as above. Integration of the HA-tag repair construct was verified with primers 32 and 33.

## ROP1 immunofluorescence assays

HFFs were grown to confluence in an 8-well μ-slide (Ibidi) and infected with *T. gondii* strains for 24 h. The slides were fixed with 4% w/v formaldehyde (Sigma) in phosphate-buffered saline (PBS) (Sigma). The cells were permeabilised with 0.2% v/v Triton X-100 (Sigma) for 15 min or 1 min and blocked with 2% w/v bovine serum albumin (Sigma) for 1 h. The cells were stained with 1:500 rat anti-HA (Roche #11867423001), followed by 1:1000 goat anti-rat 594 (Invitrogen #A11007), followed by a mixture of 1:500 mouse anti-ROP1 (Abnova #MAB17504) and 1:1000 rabbit anti-T. gondii (Abcam #ab138698), and finally with a mixture of 1:1000 goat anti-mouse 488 (Invitrogen #A11029), 1:1000 goat anti-rabbit 647 (Invitrogen #A21244), and 5 μg/mL DAPI (Sigma), each for 1h at room temperature. Images were acquired on a Nikon Ti-E inverted widefield fluorescence microscope with a Nikon CFI APO TIRF 100x/1.49 objective and Hamamatsu C11440 ORCA Flash 4.0 camera running NIS Elements (Nikon).

## ROP1 Western blotting

Parasites were purified from host cell material by syringe-lysis, filtering and washing in PBS, then lysed in RIPA buffer (Pierce) supplemented with 2x cOmplete Mini EDTA-free Protease Inhibitor Cocktail (Roche). 10 μg protein per sample was boiled for 5 min in sample loading buffer and separated by SDS-PAGE using the Mini-PROTEAN electrophoresis system (Bio-Rad). Proteins were transferred to a nitrocellulose membrane using the Trans-Blot Turbo transfer system (Bio-Rad), blocked in 2% w/v skim milk powder, 0.1% v/v Tween 20 in PBS for 1 h at room temperature, then incubated with primary antibodies in blocking buffer overnight at 4˚C. Primary antibodies used were 1:1000 mouse anti-ROP1 (Abnova #MAB17504),

1:1000 rat anti-HA (Roche #11867423001), and 1:200 mouse anti-*T. gondii* (Santa Cruz #SC-52255). Blots were stained with secondary antibodies for 1 h at room temperature: 1:10,000 goat anti-rat IRDye 680LT (Li-Cor #925–68029) and 1:10,000 goat anti-mouse IRDye 800CW (Li-Cor #925–32210). Blots were visualised using an Odyssey CLx scanner (Li-Cor).

## Plaque assays

100 parasites were inoculated onto a T25 flask of confluent HFFs and left undisturbed for seven days, following which the cells were stained with 0.5% w/v crystal violet (Sigma), 0.9% w/v ammonium oxalate (Sigma), 20% v/v methanol in distilled water.

## *In vivo* virulence assay

Parasites were syringe-lysed from intact vacuoles, passed through a 5 μm filter and resuspended in PBS. Five C57BL/6J mice per *T. gondii* strain were randomly assigned to each experimental group and intraperitoneally injected with 50,000 parasites. Parasite viability was determined by plaque assay using inocula left over after the mouse injections to be 40% for all three parasite strains used, resulting in an infective dose of 20,000 parasites per mouse. Mice were monitored daily for 30 days and euthanised if the humane endpoint was reached. All surviving mice were checked by serum antibody ELISA as previously described to confirm anti-*T. gondii* seroconversion [15].

## IFNγ restriction assays

For BMDMs, 75,000 cells per well were seeded in a 96-well μ-plate (Ibidi). For THP-1-derived macrophages, 75,000 THP-1 monocytes were seeded per well and differentiated into macrophages as above. For MEFs, 10,000 cells were seeded per well and grown to confluence prior to infection. BMDMs and MEFs were stimulated with 10 ng/mL (~100 U/mL) recombinant mouse IFNγ (Gibco) for 24 h prior to infection or left unstimulated. THP-1-derived macrophages were stimulated with 50 ng/mL (~100 U/mL) recombinant human IFNγ (BioTechne) for 24 h prior to infection or left unstimulated. The plates were infected with parasite lines at an MOI of 0.3 for 24 h, with at least three wells for each line with and without IFNγ. The plates were fixed with 4% w/v formaldehyde for 15 min and stained with 5 μg/mL DAPI and 5 μg/mL CellMask deep red plasma membrane stain (Invitrogen) for 1 h at room temperature. Biological replicates were carried out on different days with independently prepared host cells.

The plates were imaged on an Opera Phenix High-Content Screening System (PerkinElmer) with a 40x/1.1 NA water immersion objective. 25 fields of view with 3–5 focal planes (depending on the host cell type) were imaged per well. Analysis was performed in Harmony v5 (PerkinElmer) on a maximum projection of the planes. Image acquisition parameters and analysis sequence are detailed in S7 Data. For each well, the total number of host cell nuclei and *T. gondii* vacuoles in the captured fields of view was determined by thresholding on the DAPI and mCherry signal. The number of parasite nuclei in each vacuole was determined based on DAPI signal to define the total number of parasites the the captured fields of view and the mean number of parasites per vacuole in each well. A pseudo-count equivalent to one vacuole containing four parasites (the BMDM dataset-wide mean vacuole size) was added to each well in the BMDM data as no parasites were detected in some IFNγ-stimulated wells infected with PRUΔGRA12. For each *T. gondii* line, IFNγ-mediated restriction was calculated as the median tachyzoite number/vacuole number/vacuole size/host cell number in the IFNγ-stimulated wells as a percentage of the median in the unstimulated wells. Differences between strains were tested by paired two-sided *t*-test with Benjamini-Hochberg adjustment.

## Propidium iodide uptake assay

60,000 BMDMs per well of a Falcon black-walled, clear-bottom 96-well plate were stimulated with IFNγ for 24 h as described above. The plates were infected with parasite lines at an MOI of 0.3 or 3 at the same time as propidium iodide (Invitrogen) was added to a concentration of 5 μg/mL. Images of each well were acquired every 30 minutes between 1 h and 12 h post-infection on a Nikon Ti-E inverted widefield fluorescence microscope maintained at 37°C with a Nikon CFI Plan Fluor 4x/0.13 objective and Hamamatsu C11440 ORCA Flash 4.0 camera running NIS Elements (Nikon). After 12 h, Triton X-100 was added to a concentration of 1% v/v to fully permeabilise the cells and a final image was captured of each well. In each image, the total fluorescence signal was measured. The percentage of propidium iodide uptake in each well at each timepoint was calculated by subtracting the first measurement at 1 hpi to remove background fluorescence signal and normalising the total fluorescence intensity following full permeabilisation to 100% uptake. The mean propidium iodide uptake across three technical replicates for each strain was taken to represent each biological replicate. Differences between strains were tested by paired two-sided *t*-test with Benjamini-Hochberg adjustment.

## Transmission electron microscopy

Confluent HFFs grown on glass coverslips were infected with *T. gondii* lines for 24 h, fixed with 2.5% glutaraldehyde, 4% formaldehyde in 0.1 M phosphate buffer (PB) for 30 min and transferred to a BioWave Pro+ microwave for processing (Pelco; Agar Scientific). The microwave program used is detailed in **S8 Data**. The cells were washed with PB twice on the bench and twice in the microwave 250 W for 40 s, stained with 1% reduced osmium for 14 min under vacuum (with/without 100 W power at 2 min intervals), and then washed twice on the bench and twice in the microwave with PB. A further stain with 1% tannic acid for 14 min (with/without 100 W power at 2 min intervals under vacuum) was followed by a quench with 1% sodium sulfate at 250 W for 2 min under vacuum and bench and microwave washes in water (as for PB). The blocks were then dehydrated in a graded ethanol series of 70%, 90%, and 100%, each performed twice at 250 W for 40 s. Exchange into Epon resin (Taab Embed 812) was performed with 50% resin in ethanol, followed by three 100% resin steps, each at 250 W for 3 min, with 30 s vacuum cycling. Finally, the samples were baked for 24 h at 60°C. 80 nm sections were stained with lead citrate and imaged in a JEM-1400 FLASH transmission electron microscope (JEOL).

## Rhoptry secretion assays

**ROP16-mediated phosphorylation of STAT6.**   T25 flasks of confluent HFFs were infected with 1 million parasites for 2 h, after which the HFFs were dissociated and fixed with methanol for 10 minutes. The cells were stained with 1:200 rabbit anti-phospho-STAT6 (Cell Signaling #56554) and 1:200 mouse anti-*T. gondii* (Santa Cruz #SC-52255) overnight at 4°C, followed by 1:1000 goat anti-rabbit 488 (Invitrogen #A11008), 1:1000 goat anti-mouse 594 (Invitrogen #A11005), and 5 μg/mL DAPI for 1 h at room temperature. Data were collected on an LSR II flow cytometer (BD) running FACSDiva v9 (BD) and analysed with FlowJo v10 (www.flowjo.com). The median anti-phospho-STAT6 signal in the infected cells was determined for each sample, and the median technical replicate taken to represent the biological replicate. The data were scaled to RHΔKU80 = 1 AU and differences between strains tested by two-sided Wilcoxon rank sum test with Benjamini-Hochberg adjustment.

**ROP17-dependent induction of cMyc.**   HFFs were grown to confluence in an 8-well μ-slide (Ibidi) and serum starved for 24 h before infection in 0.1% FBS medium. Each well was infected with 40,000 parasites for 24 h in 0.1% FBS medium before fixation with 4% w/v

formaldehyde for 15 min, permeabilisation with 0.2% v/v Triton X-100 for 15 min, and blocking with 2% w/v BSA for 1 h. The cells were stained with 1:800 rabbit anti-cMyc (Cell Signaling #5605) and 1:200 mouse anti-*T. gondii* (Santa Cruz #SC-52255) for 2 h at room temperature, followed by 1:1000 goat anti-rabbit 488 (Invitrogen #A11008), 1:1000 goat anti-mouse 594 (Invitrogen #A11005), and 5 µg/mL DAPI for 1 h at room temperature. Images were acquired on a Nikon Ti-E inverted widefield fluorescence microscope with a Nikon Plan APO 40x/0.95 objective and Hamamatsu C11440 ORCA Flash 4.0 camera running NIS Elements (Nikon) and analysed using ImageJ [63]. The median cMyc fluorescence intensity in each nucleus was determined and the median nucleus taken as representative of a replicate. The median background cMyc fluorescence intensity was subtracted, and the data normalised to RHΔKU80 = 1 AU for each biological replicate. Differences between strains were tested by two-sided Wilcoxon rank sum test with Benjamini-Hochberg adjustment.

**ROP18-dependent inhibition of IRGB6 recruitment.**   75,000 BMDMs per well were seeded in a 96-well µ-plate (Ibidi) and stimulated with 10 ng/mL (~100 U/mL) recombinant mouse IFNγ (Gibco) for 24 h prior to infection. The BMDMs were infected with parasite strains with an MOI of 0.3 for 1 h, fixed with 4% w/v formaldehyde for 15 min, permeabilised with 0.1% w/v saponin (Sigma) for 15 minutes and blocked with 2% w/v BSA for 1 h. The plate was stained with 1:4000 rabbit anti-IRGB6 (a gift from the lab of Jonathan Howard) for 1 h at room temperature, followed by 1:1000 goat anti-rabbit 488 (Invitrogen #A11008), 5 µg/mL DAPI, and 5 µg/mL CellMask Deep Red plasma membrane stain for 1 h at room temperature. Images were acquired on an Opera Phenix High-Content Screening System (PerkinElmer) as above, and analysed in Harmony v5. Image acquisition parameters and analysis sequence are detailed in S7 Data. Vacuoles were counted as recruited if the median anti-IRGB6 intensity in a 6 pixel-wide ring around the vacuole (defined by parasite-expressed mCherry signal) was more than 2.3–2.6x (depending on the maximum signal intensity in the replicate) higher than the median anti-IRGB6 signal in the rest of the infected cell. For each well the % IRGB6-recruited vacuoles was determined, and the median % recruitment per strain taken as representative of a biological replicate. Differences between strains were determined by paired two-sided *t*-test with Benjamini-Hochberg adjustment.

## Co-immunoprecipitation

Primary MEFs/HFFs were grown to confluence in T175 flasks and infected with 5 million parasites per flask for 24 h. The flasks were washed twice with chilled PBS and lysed in IP buffer on ice (50 mM Tris, 150 mM NaCl, 0.2% v/v Triton X-100, 2x cOmplete Mini EDTA-free Protease Inhibitor Cocktail, pH 7.5).

**Co-immunoprecipitation-mass spectrometry.**   For mass spectrometry analysis, the samples were lysed in 1 mL IP buffer and incubated with 40 uL per sample of Pierce anti-HA agarose matrix (Thermo) overnight at 4˚C, following which the matrix was washed three times with IP buffer and the bound proteins eluted with 30 µL 3x Sample Loading Buffer (NEB) at 95˚C for 10 min. Samples were prepared for LC-MS/MS analysis by in-gel tryptic digestion. Briefly, the eluted proteins were run 1 cm into a NuPAGE 10% Bis-Tris gel (Invitrogen) and stained with Coomassie Brilliant Blue. The gel was cut into 1 mm cubes, destained using 50% ethanol, 50 mM ammonium bicarbonate, and dehydrated with 100% ethanol. Proteins were then simultaneously reduced and alkylated with 10 mM tris(2-carboxyethyl)phosphine and 40 mM chloroacteamide in water at 70˚C for 5 min. The gel cubes were washed in 50% ethanol, 50 mM ammonium bicarbonate and dehydrated as above. Proteins were digested with 250 ng of mass spectrometry-grade trypsin (Thermo) in 50 mM HEPES, pH 8, at 37˚C overnight. Peptides were extracted from the gel into acetonitrile and dried by vacuum centrifugation.

Digested samples were solubilised in 0.1% formic acid and loaded onto Evotips (Evosep), according to the manufacturer's instructions. Following a wash with aqueous acidic buffer (0.1% formic acid in water), samples were loaded onto an Evosep One system coupled to an Orbitrap Fusion Lumos (ThermoFisher Scientific). The Evosep One was fitted with a 15 cm column (PepSep) and a predefined gradient for a 44-minute method was employed. The Orbitrap Lumos was operated in data-dependent mode (1 second cycle time), acquiring IT HCD MS/MS scans in rapid mode after an OT MS1 survey scan (R = 60,000). The MS1 target was 4E5 ions whereas the MS2 target was 1E4 ions. The maximum ion injection time utilized for MS2 scans was 300 ms, the HCD normalized collision energy was set at 32 and the dynamic exclusion was set at 15 seconds.

Acquired raw files were processed with MaxQuant v1.5.2.8 [64]. Peptides were identified from the MS/MS spectra searched against *Toxoplasma gondii* (ToxoDB) and *Mus musculus* (UniProt) proteomes using the Andromeda search engine [65]. Methionine oxidation, acetylation (N-term), and deamidation (NQ) were selected as variable modifications whereas cysteine carbamidomethylation was selected as a fixed modification. The enzyme specificity was set to trypsin with a maximum of two missed cleavages. The precursor mass tolerance was set to 20 ppm for the first search (used for mass re-calibration) and to 4.5 ppm for the main search. The datasets were filtered on posterior error probability (PEP) to achieve a 1% false discovery rate on protein, peptide and site level. Other parameters were used as pre-set in the software. "Unique and razor peptides" mode was selected to allow identification and quantification of proteins in groups (razor peptides are uniquely assigned to protein groups and not to individual proteins). Intensity-based absolute quantification (iBAQ) in MaxQuant was performed using a built-in quantification algorithm [64] enabling the "Match between runs" option (time window 0.7 minutes) within replicates.

MaxQuant output files were processed with Perseus, v1.5.0.9 [66] and Microsoft Office Excel 2016 (**S4 Data**). Data were filtered to remove contaminants, protein IDs originating from reverse decoy sequences and only identified by site. iBAQ intensities and the total intensity were $\log_2$ and $\log_{10}$ transformed, respectively. Samples were grouped according to sample type (PRUΔKU80 or ROP1-HA) and the iBAQ intensities were filtered for the presence of two valid values in at least one group. Next, missing values were imputed from the normal distribution in order to generate $\log_2$ fold-changes (L2FCs) between tested conditions and perform statistical analysis (Welch's *t*-test, $p < 0.05$, $-3 > L2FC > 3$). The L2FC threshold was set at three times the median absolute deviation.

The mass spectrometry proteomics data have been deposited to the ProteomeXchange Consortium via the PRIDE [67] partner repository with the dataset identifier PXD032319.

**Co-immunoprecipitation-Western blot.** For Western blotting analysis, the samples were lysed in 300 μL IP buffer and incubated with 30 μL Pierce anti-HA magnetic beads (Thermo) overnight at 4˚C, following which the beads were washed three times with IP buffer and the bound proteins eluted with 30 μL 3x Sample Loading Buffer (NEB) at 95˚C for 10 min. 5% of the post-IP lysate supernatant and 15% of the immunoprecipitate were separated by SDS-PAGE and transferred to a nitrocellulose membrane as above. The membrane was blocked with 2% w/v skim milk powder, 0.1% v/v Tween 20 in PBS for 1 h at room temperature, then incubated with primary antibodies in blocking buffer overnight at 4˚C. Primary antibodies used were: 1:1000 rat anti-HA (Roche #11867423001), 1:1000 mouse anti-ROP1 (Abnova #MAB17504), 1:1000 rabbit anti-C1QBP (Abcam #ab270032), 1:10,000 rabbit anti-GAPDH (Proteintech #10494-1-AP), and 1:1000 rabbit anti-GRA29 [15]. Blots were stained with secondary antibodies for 1 h at room temperature: 1:10,000 goat anti-mouse IRDye 680LT (Li-Cor #925–68020), 1:10,000 goat anti-rat IRDye 800CW (Li-Cor #925–32219), 1:10,000 donkey anti-rabbit IRDye 680LT (Li-Cor #925–68023), and 1:10,000 donkey anti-

rabbit IRDye 800CW (Li-Cor #925–32213). Blots were visualised using an Odyssey CLx scanner (Li-Cor).

## Vacuole ubiquitination assay

150,000 BMDMs per well were seeded in an 8-well μ-slide and stimulated with 10 ng/mL (~100 U/mL) recombinant mouse IFNγ for 24 h prior to infection. The BMDMs were infected with parasite strains at an MOI of 0.3 for 3 h, then washed and fixed with 4% w/v formaldehyde for 15 min. Prior to permeabilisation, the cells were blocked with 2% w/v BSA for 1 h and extracellular parasites were stained with 1:1000 rabbit anti-*T. gondii* (Abcam #ab138698) for 1 h at room temperature followed by 1:1000 goat anti-rabbit 405 (Invitrogen #A31556) for 1 h at room temperature. The cells were then permeabilsed with 0.2% v/v Triton X-100 for 15 minutes, blocked again with 2% w/v BSA for 1 h, stained with 1:200 mouse anti-ubiquitinylated proteins (Sigma #04–263) overnight at 4˚C, and finally stained with 1:1000 goat anti-mouse 488 (Invitrogen #A11029) for 1 h at room temperature. Nine tiled fields of view were captured for each well on a Nikon Ti-E inverted widefield fluorescence microscope as above. The images were blinded, and the percentage of ubiquitinated vacuoles was determined manually using ImageJ, excluding *T. gondii* cells which were positive for extracellular staining. A median of 290 vacuoles were analysed per strain per replicate. Differences between strains were determined by two-sided *t*-test with Benjamini-Hochberg adjustment.

## C1QBP immunofluorescence assays

Where MitoTracker staining was used, cells were stained with 200 nM MitoTracker Red CMXRos (Invitrogen) in complete medium for 30 min prior to fixation. The cells were fixed with 4% w/v formaldehyde in PBS, permeabilised with 0.2% v/v Triton X-100 for 15 min or 1 min, and blocked with 2% w/v bovine serum albumin for 1 h. The cells were stained with 1:100 rabbit anti-C1QBP (Abcam #ab270032) overnight at 4˚C followed by a mixture of 1:1000 goat anti-rabbit 488 (Invitrogen #A11008) and 5 μg/mL DAPI for 1 h at room temperature. Where cells were infected and MitoTracker staining was not used, the cells were additionally stained with 1:500 mouse anti-ROP1 (Abnova #MAB17504) overnight at 4˚C followed by 1:1000 goat anti-mouse 594 (Invitrogen #A11005) for 1 h at room temperature. Images were acquired on a Nikon Ti-E inverted widefield fluorescence microscope with a Nikon CFI APO TIRF 100x/1.49 objective and Hamamatsu C11440 ORCA Flash 4.0 camera running NIS Elements (Nikon).

## Supporting information

**S1 Fig. Comparison of CRISPR screen phenotypes to published datasets. A.** Correlation between *in vivo* L2FCs from this study and *in vivo* L2FCs from [15]. *r* = Pearson's product-moment correlation coefficient. **B, C, D.** Correlation between *in vitro* L2FCs from this study and **B** *in vitro* L2FCs from [15], **C** HFF passage 3 L2FCs from [18], and **D** HFF passage 8 L2FCs from [19]. *r* = Pearson's product-moment correlation coefficient. **E.** Scatter plot of median L2FCs for each gene *in vitro* and *in vivo*. Control genes for which knockouts have previously been tested for an effect on virulence in Type II strains of *T. gondii* are labelled. Genes which have been found to be essential in Type I strains of *T. gondii* are also labelled. The grey line indicates equal *in vitro* and *in vivo* L2FCs. **F.** Z-score-transformed HFF L2FCs from [18] and [19] for genes screened in this study. The grey line indicates equal z-scores in both studies and dotted lines indicate z-scores of -1 in each study.
(TIF)

**S2 Fig. Verification of *T. gondii* knockout cell lines. A.** Verification of correct integration of knockout and complementation constructs by diagnostic PCR and verification of strain genotype by restriction fragment length polymorphism (RFLP) of the SAG3 gene [61]. Knockouts were obtained by integration of an mCherry-T2A-HXGPRT linear PCR cassette facilitated by co-transfection with a Cas9-sgRNA plasmid targeting the gene of interest. For ROP1 complementation, the ROP1 coding sequence and native promoter were cloned from RHΔKU80 or PRUΔKU80 genomic DNA into the pUPRT vector, adding a single C-terminal HA tag, linearised and integrated by double homologous recombination following co-transfection with a Cas9-sgRNA plasmid targeting the UPRT locus. **B.** Verification of ROP1 and ROP1-HA expression by Western blot. **C.** Immunofluorescence verification of ROP1 knockout and complemented *T. gondii* cell lines using 1 minute permeabilisation. Scale bar = 10 μm. **D.** Plaques formed by RHΔKU80, RHΔROP1, PRUΔKU80 and PRUΔROP1 parasites after seven days' growth in a monolayer of HFFs. Scale bar = 1 cm.
(TIF)

**S3 Fig. ROP1 does not affect vacuole size or host cell death. A, B.** IFNγ-dependent growth restriction of *T. gondii* in BMDMs. BMDMs were stimulated with IFNγ for 24 h, infected with *T. gondii* cell lines for a further 24 h and parasite growth quantified by automated fluorescence imaging and analysis. **A** *T. gondii* vacuole size (mean parasites per vacuole) in IFNγ-stimulated BMDMs is shown as a percentage of the size in unstimulated BMDMs. **B** The number of IFNγ-stimulated host BMDM nuclei is shown as a percentage of the number of unstimulated BMDM nuclei. p-values were calculated by paired two-sided *t*-test with Benjamini-Hochberg adjustment. **C, D.** IFNγ-dependent growth restriction of *T. gondii* in THP-1-derived macrophages. Differentiated THP-1 macrophages were stimulated with IFNγ, infected, and parasite growth quantified as above. **C** *T. gondii* vacuole size (mean parasites per vacuole) in IFNγ-stimulated THP-1 macrophages is shown as a percentage of the size in unstimulated macrophages. **D** The number of IFNγ-stimulated host THP-1 macrophage nuclei is shown as a percentage of the number of unstimulated macrophage nuclei. p-values were calculated by paired two-sided *t*-test with Benjamini-Hochberg adjustment.
(TIF)

**S4 Fig. Propidium iodide uptake assays for parasite-induced host cell death. A, B.** Propidium iodide uptake of IFNγ-stimulated BMDMs infected at an MOI of **A** 0.3 or **B** 3. Propidium iodide fluorescence was measured every 30 minutes from 1–12 hours post-infection. Curves represent the mean of five replicates. Uptake at 12 hours post-infection is shown in Fig 3.
(TIF)

**S5 Fig. Additional rhoptry TEM images. White arrowheads indicate rhoptries.** Scale bar = 500 μm.
(TIF)

**S6 Fig. Verification of *T. gondii* C-terminal tagged cell lines by PCR and Western blot. A.** Verification of correct integration of C-terminal HA-tagging construct by diagnostic PCR and verification of strain genotype by restriction fragment length polymorphism (RFLP) of the SAG3 gene [61]. HA-tagged cell lines were obtained by double homologous recombination with an HA-HXGPRT linear PCR cassette facilitated by co-transfection with a Cas9-sgRNA plasmid targeting the 3' UTR of ROP1. Clonal *T. gondii* cell lines were obtained by limiting dilution. **B.** Verification of ROP1 and ROP1-HA expression by Western blot.
(TIF)

**S7 Fig. ROP1 does not affect vacuole ubiquitination. Percentage of ubiquitinated vacuoles.** BMDMs were stimulated with 100 U/mL IFNγ for 24 h, infected for 3 h, fixed, and stained with an anti-ubiquitinylated proteins antibody. The percentage of ubiquitinated vacuoles was quantified manually from blinded immunofluorescence microscopy images. p-values were calculated by paired two-sided *t*-test with Benjamini-Hochberg adjustment.
(TIF)

**S8 Fig. Immunofluorescence localisation of C1QBP. A, B, C.** Immunofluorescence localisation of C1QBP in **A** C57BL/6J BMDMs stimulated +/- 100 U/mL IFNγ for 24 h, **B** primary C57BL/6J MEFs, and **C** HFFs. Scale bars = 10 μm.
(TIF)

**S9 Fig. C1QBP^flox/flox (WT) and C1QBP^-/- (KO) MEFs do not restrict *T. gondii* growth. A.** Validation of C1QBP^flox/flox and C1QBP^-/- immortalised MEFs by immunofluorescence assay. Scale bar = 10 μm. **B.** Validation of C1QBP^flox/flox and C1QBP^-/- immortalised MEFs by Western blot. **C, D, E, F.** IFNγ-dependent growth restriction of *T. gondii* in C1QBP^flox/flox (WT) and C1QBP^-/- (KO) immortalised MEFs. MEFs were stimulated with IFNγ for 24 h, infected with *T. gondii* cell lines for a further 24 h, and parasite growth quantified by automated fluorescence imaging and analysis. Parasite growth in IFNγ-stimulated BMDMs is shown as a percentage of that in unstimulated BMDMs in terms of **C** total number of *T. gondii* parasites, **D** number of *T. gondii* vacuoles, **E** vacuole size, and **E** number of host cells. p-values were calculated by paired two-sided *t*-test with Benjamini-Hochberg adjustment.
(TIF)

**S1 Data. CRISPR knockout screen results. A.** Raw protospacer sequencing read counts. **B.** Normalised protospacer sequencing read counts. **C.** Protospacer L2FCs. **D.** Gene L2FCs and p-values, and DISCO scores. **E.** Comparison of L2FCs to [15], [19], [16], and [18].
(XLSX)

**S2 Data. BMDM IFNγ restriction assay results. A.** *T. gondii* cell number, vacuole number, mean vacuole size and host cell number per well. **B.** Median *T. gondii* cell number, vacuole number, mean vacuole size, host cell number and survival percentage +/- IFNγ per strain per replicate.
(XLSX)

**S3 Data. THP-1 IFNγ restriction assay results. A.** *T. gondii* cell number, vacuole number, mean vacuole size and host cell number per well. **B.** Median *T. gondii* cell number, vacuole number, mean vacuole size, host cell number and survival percentage +/- IFNγ per strain per replicate.
(XLSX)

**S4 Data. Co-immunoprecipitation mass spectrometry results.**
(XLSX)

**S5 Data. MEF FNγ restriction assay results. A.** *T. gondii* cell number, vacuole number, mean vacuole size and host cell number per well. **B.** Median *T. gondii* cell number, vacuole number, mean vacuole size, host cell number and survival percentage +/- IFNγ per strain per replicate. **C.** Ratio of survival in C1QBP^-/- vs. C1QBP^flox/flox MEFs.
(XLSX)

**S6 Data. Primers sequences used in this work.**
(XLSX)

**S7 Data. Opera Phenix image acquisition parameters and Harmony analysis sequence.**
(PDF)

**S8 Data. Microwave program used for TEM sample preparation.**
(XLSX)

## Acknowledgments

We thank all members of the Treeck laboratory for critical discussions. We thank Rachael Instrell and Becky Saunders (High-Throughput Screening Science Technology Platform, The Francis Crick Institute, London, United Kingdom) for assistance with sgRNA library preparation. We thank Takeshi Uchiumi and Dongchon Kang (Kyushu University, Fukuoka, Japan) for providing the C1QBP<sup>flox/flox</sup> and C1QBP<sup>-/-</sup> MEFs. We thank members of the Advanced Sequencing Facility, Biological Research Facility, and Cell Services Science Technology Platforms at the Francis Crick Institute for support. We thank VEuPathDB [68] for providing access to genomic and other large-scale datasets for *T. gondii*.

## Author Contributions

**Conceptualization:** Simon Butterworth, Jeanette Wagener, Joanna C. Young, Moritz Treeck.

**Formal analysis:** Simon Butterworth, Malgorzata Broncel.

**Funding acquisition:** Moritz Treeck.

**Investigation:** Simon Butterworth, Francesca Torelli, Eloise J. Lockyer, Jeanette Wagener, Ok-Ryul Song, Malgorzata Broncel, Matt R. G. Russell, Aline Cristina A. Moreira-Souza, Joanna C. Young.

**Project administration:** Moritz Treeck.

**Supervision:** Joanna C. Young, Moritz Treeck.

**Visualization:** Simon Butterworth.

**Writing – original draft:** Simon Butterworth, Moritz Treeck.

**Writing – review & editing:** Simon Butterworth, Francesca Torelli, Eloise J. Lockyer, Jeanette Wagener, Ok-Ryul Song, Malgorzata Broncel, Matt R. G. Russell, Aline Cristina A. Moreira-Souza, Joanna C. Young, Moritz Treeck.

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
