## [Decision Letter · Decision Letter 0]

5 Nov 2022

Dear Dr. Treeck,

Thank you very much for submitting your manuscript "Toxoplasma gondii virulence factor ROP1 reduces parasite susceptibility to murine and human innate immune restriction" for consideration at PLOS Pathogens. As with all papers reviewed by the journal, your manuscript was reviewed by members of the editorial board and by several independent reviewers. The reviewers appreciated the attention to an important topic. Based on the reviews, we are likely to accept this manuscript for publication, providing that you modify the manuscript according to the review recommendations. We would like to invite the resubmission of a slightly revised version that takes into account the reviewers' comments regarding the results/figure section of your manuscript. No additional experiments are required.

Sincerely,

Philipp Olias

Associate Editor

PLOS Pathogens

Dominique Soldati-Favre

Section Editor

PLOS Pathogens

Kasturi Haldar

Editor-in-Chief

PLOS Pathogens

orcid.org/0000-0001-5065-158X

Michael Malim

Editor-in-Chief

PLOS Pathogens

orcid.org/0000-0002-7699-2064

Reviewer Comments (if any, and for reference):

Reviewer's Responses to Questions

**Part I - Summary**

Reviewer #1: The authors have addressed concerns raised by the previous reviewers by addition of new data and revision of the manuscript. I have no further concerns and feel this will be an important contribution for the community.

Reviewer #2: In this manuscript, the authors use a CRISPR screen of secreted factors to identify those that are important for immune evasion in the host. In addition to a number of known factors and an array of intriguing candidates, they identify ROP1 as an important regulator of both the human and murine immune response. They then demonstrate that the protein associates with host C1QBP, providing a potential mechanism of action. The authors have responded well to the comments from the previous review - specific comments are below.

Reviewer #3: In the manuscript “Toxoplasma gondii virulence factor ROP1 reduces parasite susceptibility to murine and human innate immune restriction” by Butterworth et al. a CRISPR screen was performed to identify novel Toxoplasma genes that determine in vivo virulence. This is a repeat of the in vivo screen the Treeck lab performed in their 2019 Nature Communications paper. However, in this screen 75 genes that were not previously assessed by the Treeck lab (61 genes when comparing previous Treeck lab and Saeij lab in vivo screen data) were added.

From this screen novel in vivo virulence candidates were identified that were not previously studied in vivo (but see Major Comment 1). Of these ROP1 was further studied and shown to determine parasite resistance to IFNy in both murine and human cells. ROP1 immunoprecipitation data indicate ROP1 can interact with C1QBP. Interestingly, there were significant differences in parasite growth between C1QBPflox/flox and C1QBP-/- MEFs which were dependent on the presence of ROP1 in the parasites. This indicates that the ROP1-C1QBP interaction is relevant for parasite growth. However, confusingly IFNy did not restrict Toxoplasma in the C1QBPflox/flox MEFs. In my opinion this is likely due to the fact that these MEFs were immortalized. We have seen that immortalization of HFFs and MEFs strongly affects the IFNy restriction phenotype.

Overall, this is an interesting manuscript that shows ROP1 is a novel Toxoplasma virulence factor that likely functions through its interaction with C1QBP. A more definitive role of for the ROP1-C1QBP interaction in IFNy-mediated parasite restriction will likely require knocking out C1QBP in non-immortalized cells.

**Part II – Major Issues: Key Experiments Required for Acceptance**

Reviewer #1: (No Response)

Reviewer #2: None, major issues previously raised were addressed in this submission

Reviewer #3: No new experiments are required. However, I recommend they rewrite the result section of "in vivo virulence genes" and remake figure 1B (see Major Comment 1).

Major Comments:

1) The main goal of this screen was to identify Toxoplasma genes that determine fitness in vivo but not in vitro. However, many of the genes that are claimed to be in vivo fitness genes already have large fitness effects in vitro based on data from genome-wide CRISPR screens performed by other labs (Lourido lab (Sidik et al 2016) and Saeij lab (Wang et al. 2020). Although the in vitro scores have a high Pearson correlation with previously published in vitro fitness scores (e.g., Pearson correlation with Sidik et al. is 0.54 indicating the reproducibility with previous CRISPR screens) the actual in vitro fitness scores are compressed and all scores are between -0.5 and 0.5. This indicates that the sgRNAs generated mutant parasites are not depleted in vitro after the 5 days of passaging. This is likely because of the smaller library size compared to the genome-wide libraries used in previous screens. However, this leads to the artificial identification of “in vivo” fitness genes that in fact already have a strong phenotype in vitro and are therefore not in vivo fitness genes but in vitro fitness genes.

A good example of this is GRA1, which is an essential gene (Rommereim et al. 2016). However, in this manuscript it is shown with an in vitro fitness of -0.2 while in Sidik et al. it was -5.3 and in Wang et al it was -6.6 (note that these are Log2FC scores and therefore the difference with -0.2 is extremely large!). Similarly, RASP2, GRA39, 205190, 301270, CLPTM1 and some others are genes already have significant in vitro fitness defects (see Sidik et al. and Wang et al.). When reanalyzing their data, now using the Sidik et al. passage 3 in vitro scores, and the Wang et al. passage 8 in vitro scores, it seems now only GRA2, GRA4, GRA25, GRA5, ROP1, GRA45, CDPK3, ROP18, and GRA12 are putative in vivo virulence genes (in vivo score <-1 in this screen and score >-1.5 in Sidik et al . passage 3 and Wang et al. passage 8). Of these only ROP1, GRA45 and CDPK3 were not previously included in the in vivo CRISPR screens of Young et al. and Sangare et al. CDPK3 (Wu et al 2022) and GRA45 (Wang et al. 2020) have already been studied in vivo. Therefore, ROP1 is the only novel identified gene from this screen. Overall, I suggest they remake Figure 1B using previously published in vitro fitness scores and rewrite the section discussing the in vivo CRISPR screen hits. Preferably they would use the Wang et al. in vitro passage 8 scores as the additional 5 days in vivo should be compared to an additional 5 days in vitro (which might correspond better to the Wang et al. in vitro passage 8).

2) The authors state “Seven out of 150 unambiguous orthologues present in both screens had L2FCs of less than -1 in both screens, indicating that they are important for immune evasion of both the RH and PRU strains.” This not entirely correct. ROP1 had a naïve BMDM fitness score of 1.13 and an IFNy fitness score of -0.17 in the Wang et al. screen. Although it is correct that the difference between naïve and IFNy fitness <-1 because the IFNy fitness score (which was IFNy vs. plasmid input) is only -0.17, ROP1 was not identified as important for fitness in IFNy-stimulated BMDM because the fitness score in IFNy-stimulated BMDMs was >-1. Because of that it was not listed as a hit in the Wang et al. paper. Furthermore, the P-value was also not significant.

3) The claim that ROP1 prevents vacuole disruption is not substantiated by data. It is also possible it affects early parasite egress. In fact, the host cell death with high MOI seems to be consistent with early parasite egress.

**Part III – Minor Issues: Editorial and Data Presentation Modifications**

Reviewer #1: (No Response)

Reviewer #2: 1. Importantly, the authors show that PRUdeltaROP1 parasites are avirulent in vivo. However, the phenotype is not completely rescued by complementation (Fig 2C and in weight loss in 2D). In addition, complementation is not addressed at all in the text. While this is not problematic (many knockouts show incomplete recovery in Toxo), it should be discussed in the results. Is this due to the C-terminal HA tag or subtle levels of the protein in the complemented line?

2. The link between ROP1 and C1QBP remains questionable and may be an artefact of these charged proteins interacting in lysates only – however the authors do clearly state this possibility and leave it as a candidate interactor which can be functionally resolved with additional studies

3. The authors make the statement “Since ROP1 from both strains can pull down C1QBP, we infer that this variant repeat region is likely not involved in the interaction”. I don't think that the variable number of repeats means this region is any less likely to be involved in the interaction.

Reviewer #3: Minor comments:

1) They mention “We obtained such scores for 164 genes after filtering to remove genes with fewer than three well-represented sgRNAs (Supplementary Data 1).” It was unclear to me what exactly they mean by this. If a Toxoplasma gene is highly essential in vivo wouldn’t all the sgRNAs targeting this gene disappear when the parasites are grown in vivo? Or do they mean with fewer than three well represented in the plasmid pool? Please specify.

2) I’m not sure if I am misinterpreting the mass-spec suppl data but only 5, 5, and 1 ROP1 peptides were identified in the 3 IP experiments. It seems not a lot of ROP1 was immunoprecipitated and it is a bit strange that more C1QBP is immunoprecipitated than the actual ROP1.

3) The authors seem to focus on the ROP1 PVM localization. However, I suppose ROP1 is also localized in cytoplasm when rhoptry bulb contents are secreted into the host cytoplasm. However, this might be difficult to visualize.

PLOS authors have the option to publish the peer review history of their article (what does this mean?). If published, this will include your full peer review and any attached files.

Reviewer #1: No

Reviewer #2: No

Reviewer #3: **Yes: **Jeroen P.J. Saeij

Figure Files:

Data Requirements:

Reproducibility:

References:

---

## [Editor Report · Decision Letter 1]

24 Nov 2022

Dear Dr Treeck,

We are pleased to inform you that your manuscript '*Toxoplasma gondii* virulence factor ROP1 reduces parasite susceptibility to murine and human innate immune restriction' has been provisionally accepted for publication in PLOS Pathogens.

Best regards,

Philipp Olias

Academic Editor

PLOS Pathogens

Dominique Soldati-Favre

Section Editor

PLOS Pathogens

Kasturi Haldar

Editor-in-Chief

PLOS Pathogens

orcid.org/0000-0001-5065-158X

Michael Malim

Editor-in-Chief

PLOS Pathogens

orcid.org/0000-0002-7699-2064
---

## [Editor Report · Acceptance letter]

3 Dec 2022

Dear Dr. Treeck,

We are delighted to inform you that your manuscript, "*Toxoplasma gondii* virulence factor ROP1 reduces parasite susceptibility to murine and human innate immune restriction," has been formally accepted for publication in PLOS Pathogens.

Best regards,

Kasturi Haldar

Editor-in-Chief

PLOS Pathogens

orcid.org/0000-0001-5065-158X

Michael Malim

Editor-in-Chief

PLOS Pathogens

orcid.org/0000-0002-7699-2064